# Proximal and fatty acid analysis in *Ostrea chilensis, Crassostrea gigas* and *Mytilus chilensis* (Bivalvia: Mollusca) from southern Chile

Andrea Valenzuela[1]*, Pablo A. Oyarzún[2], Jorge E. Toro[1,3], Jorge M. Navarro[1,4], Oscar Ramírez[5], Ana Farias[3,6]

1 Instituto de Ciencias Marinas y Limnológicas, Universidad Austral de Chile Independencia, Valdivia, Chile,
2 Centro de Investigación Marina Quintay (CIMARQ), Universidad Andres Bello, Quintay, Chile,
3 Interdisciplinary Network of Advanced Research for Marine Larviculture of Species with Complex Lifecycles (INLARVI), 4 Centro Fondap de Investigación Dinámica de Ecosistemas Marinos de Altas Latitudes (IDEAL), Universidad Austral de Chile, Valdivia, Chile, 5 Programa de Magister en Medio Ambiente y bioseguridad en Acuicultura, Universidad Austral de Chile, Puerto Montt, Chile, 6 Instituto de Acuicultura, Universidad Austral de Chile, Puerto Montt, Chile

* andreavalenz@gmail.com

**Data Availability Statement:** All relevant data are within the paper.

## Abstract

Oysters and blue mussels are important hydrobiological resources for aquaculture. In Chile, they are farming on the Chiloé island, where around 18% of the world's mussels are produced, however, their nutritional dynamics are largely unknown. For this reason, the objective of this study was to determine the proximal biochemical composition and the fatty acid profile in the Chilean oyster (*Ostrea chilensis*), the Pacific oyster (*Crassostrea gigas*) and the Chilean mussel (*Mytilus chilensis*), to perform an intra and interspecific comparison. Shellfish sampled in winter were characterized by a high protein content, followed by medium values for lipid content and a low carbohydrate content compared to similar species in Europe. Also, oysters and mussels were found to be rich in omega-3 long chain polyunsaturated fatty acid (n-3 LC-PUFA), so they can be considered excellent functional food option for a healthy human diet. Their high contribution of n-3 LC-PUFA ranged between 5.2–12.9 µg FA mg$^{-1}$ dry weight with high n-3/n-6 ratios, which depends on both the species and the on-growing location. Both taxa can be considered a plausible option to promote a healthy diet of marine origin in future generations. Also, these results could benefit the projection and development of aquaculture of these mollusks.

## Introduction

Proximal composition studies and fatty acid analysis make it possible to determine the nutritional value of any organism and provide relevant information to understand its metabolism and energy balance [1–3]. In bivalve mollusks, the nutritional value varies between species, including differences between males and females [4–9]. One of the relevant nutritional characteristics of bivalves is their production of omega-3 highly unsaturated fatty acids (n-3 HUFA) or long chain poly unsaturated fatty acids (n-3 LC-PUFA), especially they have high contents of eicosapentaenoic acid (20:5n-3, EPA) and docosahexaenoic acid (22:6n-3, DHA) [10]. Both fatty acids are main components of phytoplanktonic diet of bivalves. The phytoplankton are

**Funding:** Funding was obtained through the following research funds: Fondef ID19I10214 and ID16I10018.

**Competing interests:** The authors have declared that no competing interests exist.

the main producers of n-3 LC-PUFA in the marine food chain [11, 12]. [13] suggests that the EPA + DHA content of mussels, clams and oysters is usually higher than that of other bivalves. In the *Magallana bilineata* oyster (formerly *Crassostrea madrasensis*), polyunsaturated fatty acids (PUFAs) were estimated to be the highest total lipids, among which eicosapentaenoic acid (EPA), docosahexaenoic acid (DHA), and acid linoleic stand out [14].

During the last decades, research has been focused on understanding the nutritional composition of bivalves of commercial interest, mainly because they are an important source of protein, lipids, carbohydrates, minerals and essential vitamins of great value for the human population [15–17]. However, most bivalves lack the ability to synthesize PUFA from saturated precursors [18]. Therefore, the levels of these components are determined by the food intake, or by the variability of the food they consume [19]. Other factors related to the proportion of fatty acids in mollusks have also been described, such as the physiological state of the organism, growth, the organism's response to variations in the environment, and internal and external factors that affect a moment in the life cycle of each individual [2, 9, 20]. For example, investigations on marine mussels have described that the proximal composition is related to the gametogenic cycle and the availability and quality of the food [2, 21, 22]. Therefore, the cultivation area has a strong influence on the nutritional components that mollusks acquire [2, 16, 23].

Oysters and blue mussels are bivalve mollusks of aquaculture importance in Chile [24]. The Chilean oyster (*Ostrea chilensis*, Küster, 1844), the Pacific oyster (*Crassostrea gigas*, Thunberg, 1793) and the Chilean blue mussel (*Mytilus chilensis*, Hupé, 1854) inhabit the Southeast Pacific coast. *Ostrea chilensis* forms sea beds on rocky or muddy hard bottoms up to 8 m deep. It has a limited range of distribution in Chile between Chiloé Island and Guaitecas Islands [25] and also in the nearshore waters of New Zealand [26]. However, its cultivation has developed in the northern area of Chiloé Island (city of Ancud). Currently, harvesting about 400 tons per year and the demand for the product has increased during the last decade [24]. The Pacific Oyster or *Crassostrea gigas* is an exotic species introduced in Chile (Coquimbo Region) in 1978 for commercial purposes [27]. In Chile, its cultivation has been carried out mainly in the III and IV regions (26˚30S to 29˚54'S—North of Chile) to export seeds to the United States (https://fch.cl/noticia/la-ruta-de-las-semillas-de-ostra-de-cultimar/, last accesed 11/08/2020) for on-growing mainly in North of Chile (about 500 ton), while at The Lakes Region its harvesting is reduced to 11 tons at the island of Chiloé, Chile (e.g. Hueihue and Faro Corona, 42˚52'S). On the other hand, the Chilean mussel (*Mytilus chilensis*), also known as 'chorito', is distributed in the Eastern Pacific Ocean from Puerto Saavedra (Chile) to Ushuaia (Argentina) [28]. In Chile, the annual landing reaches 379,096 tons, which explained 18% of world production in 2019 that are mainly exported to Asia and Europe [24, 29]. Chile is one of the main exporters of blue mussels in the world and all the production (99.9%) comes from The Lakes Region, mainly Chiloé island. Oysters and mussels have increased their production in Chile through the last decade, however, the variability of their nutritional components is largely unknown.

This study aimed to evaluate the proximal biochemical composition and the fatty acid profile of the Chilean oyster, Japanese oyster and Chilean mussel, in order to carry out an intra and interspecific comparison of commercial bivalves from aquaculture in southern Chile, to nutritionally characterize the food quality of these hydrobiological resources.

## Material and methods

### Sampling and determination of the proximal composition and fatty acid profile of mollusks

The bivalve samples used in this study were manipulated according to the criteria on the use of animals in research granted by the Bioethics Committee of the Universidad Austral de Chile.

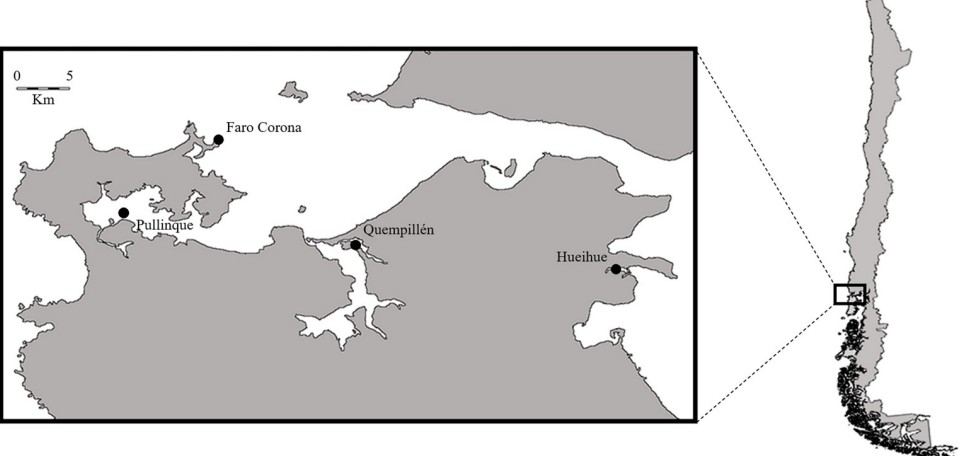

**Fig 1. Map of the collection sites of oysters (*Ostrea chilensis* and *Crassostrea gigas*) and Chilean mussels (*Mytilus chilensis*) in the north of Chiloé Island, Chile.**

The animals were collected on two occasions in the north of Chiloé Island. The first sample collection was performed in September 2017 (at the end of winter) when 60 individuals of *Ostrea chilensis* (Chilean oyster) were randomly extracted from the locality of Quempillén (41˚ 52′ 0″ S, 73˚ 49′ 60″ W), which were separated 3 groups of 20 individuals, each group was considered as a replicate. On the same occasion, 60 individuals of *Ostrea chilensis* were collected from Pullinque locality (41˚ 50' 45" S, 73˚ 56' 08" W) and 40 individuals of *Cassostrea gigas* (Pacific oyster) from Faro Corona locality (41˚ 47' 06″ S, 73˚ 53′ 18″ W), each group of 20 individuals was considered as a replicate (Fig 1). Then, in August 2018 (late winter), 60 *O. chilensis*, 60 *C. gigas* and 60 *M. chilensis* specimens were randomly collected following the same type of sampling described above, all from Hueihue locality (41˚ 54' 05" S, 73˚ 29' 28" W), north of Chiloé Island (Fig 1). The environmental conditions of temperature and salinity for each zone were: Quempillén (14.7˚C and 22.4 ppt), Pullinque (13.6˚C and 30.6 ppt), Faro Corona (14.1˚C and 29.1 ppt) and Hueihue (12.3˚C and 31.8 ppt) [30]. The samples were transferred dry, with a transfer time of 2 hours to the Marine Invertebrate Hatchery of the Aquaculture Institute of the Universidad Austral de Chile (HIM-UACh), where the tissues were processed without shells. Each sample was stored in a refrigerated/frozen Ziploc bag with the meat of 20 individuals corresponding to replicates 1, 2 and 3 of each sampling. For each sample of 20 individuals the tissues were pooled and freeze dried in a Savant freeze dryer (-80˚C), then they were ground to homogenize and stored at -41˚C until the respective biochemical analysis. The total length of bivalves from each sample was individually measured with a caliper (± 0.1 mm) and weighed on a Sartorius analytical balance (± 0.01 g).

The samples were dissected with the help of a scalpel. The relative weight index [31] based on the allometric equation between the whole wet weight vs the total length was calculated according to unpublished results, the bivalves in rearing conditions showed the allometric relationship of $WW = 0.021^*L^{2.16}$ for *M. chilensis*, $WW = 0.0004^*L^{2.77}$ for *O. chilensis*, and $WW = 0.079^*L^{1.48}$ for *C. gigas*, calculated from the same samplings of the study, where WW and L are the wet weight in g and total length in cm of the specimens. In this way, the relative weight index (Wr) for each species was calculated as $Wr = WW / [0.021^*L^{2.16}]$, $Wr = WW / [0.0004^*L^{2.77}]$, $Wr = WW / [0.079^*L^{1.48}]$, respectively.

The proximal and fatty acid compositions of tissues were determined according to [32]. Carbon, hydrogen and nitrogen content were analyzed using a CHN analyzer (LECO

CHN-900) with 1 mg sample weighed in a microbalance (± 0.001 mg; METTLER-TOLEDO XP2U). Crude protein was calculated based on the N x 6.25 and reported in percentage. The total nitrogen content of the sample was converted into a value called "crude protein" by multiplying it by 6.25, a constant based on the conventional assumption that any protein is composed of 16% nitrogen. Total lipid content was obtained gravimetrically, after extraction, through the method of [33]. Methylation and quantification of fatty acids were performed according to the method described by [34]. The fatty acid methyl esters (FAMEs) from the total lipids were analyzed in a gas chromatograph (Focus GC, Thermo Fisher Scientific Inc., Waltham, MA, USA) equipped with an autosampler. The separation was performed using hydrogen as a carrier gas in a capillary column RESTEC RT-2560 of 100 m length, 0.25 mm internal diameter and 0.2 mm phase, using a splitless injection program. The initial temperature of 140°C was maintained for 5 min before starting a ramp at a rate of 5°C min up to 240°C, held for 20 min. The detector was set at 260°C. Nonadecanoic acid (19:0) was used as an internal standard.

Ash content was obtained after calcination at 500°C for 6 h (Vulcan Muffle model A-550). The carbohydrates were calculated as the difference between 100% of the dry weight and the sum of protein, lipid, and ash. Values to calculate energy conversion were 23.7, 39.5 and 17.2 kJ g$^{-1}$ for protein, lipid and carbohydrate, respectively [35].

The nutritional status indicators of bivalve meat were calculated in dry basis as:

i. Protein, lipid, carbohydrates and ash = per (g 100 g dry weight)$^{-1}$.

ii. Energy = (g protein g dry weight$^{-1}$×23.7 kJ g$^{-1}$) + (g lipid g dry weight$^{-1}$×39.5 kJ g$^{-1}$) + (g carbohydrate g dry weight$^{-1}$x17.2 kJ g$^{-1}$) = kJ g dry weight$^{-1}$ x 1000 = MJ kg$^{-1}$ dry weight.

iii. Protein/energy ratio (P/E) = g protein per kg dry weight / MJ per kg dry weight.

iv. Carbon/Nitrogen ratio (C/N) = [g carbon g dry weight$^{-1}$/12 g] / [g nitrogen g dry weight$^{-1}$/14 g].

v. Yield = Meat wet weight x 100/ Total wet weight [36].

## Statistical analysis

The data from each sampling year were treated independently. A one-way ANOVA was performed when the assumptions of normality and homoscedasticity (Levene's test and White's test, respectively) were fulfilled in the data corresponding to fatty acids and proximal composition values. If the data did not meet the statistical assumptions, a Kruskal-Wallis rank ANOVA was performed. In those cases, where significant differences were found, a Tukey or Dunn posterior test was carried out according to the analysis of variance test used. These significant differences are indicated with different letters (Tables 1 to 4).

To analyze the differences between profiles of fatty acids of the species at different locations, a multivariate analysis of principal components (PCA) was performed using a correlation matrix, to define new uncorrelated variables (components) established from linear combinations of the original variables (Fig 2).

## Results

### Interspecific analysis

The three species analyzed (*O. chilensis*, *C. gigas* and *M. chilensis*) showed significant differences in protein content (H $_{(2, n = 9)}$ = 7.20; $P$ = 0.027). The highest value was obtained by *M*.

**Table 1. Proximal composition and condition indices of oysters and mussels from Hueihue (Chiloé Island, Chile).** Each value represents the average of three independent replicates ± standard error. The letters indicate significant differences between averages.

| | *Ostrea chilensis* | *Crassostrea gigas* | *Mytilus chilensis* |
|---|---|---|---|
| | (Hueihue) | (Hueihue) | (Hueihue) |
| **Proximal composition** | | | |
| Protein (% dry weight) | 42.81 ± 0.24[a] | 50.07 ± 0.88[b] | 52.54 ± 0.29[c] |
| Lipid (% dry weight) | 17.70 ± 1.11[ab] | 19.54 ± 0.28[b] | 14.50 ± 2.55[a] |
| Carbohydrate (% dry weight) | 20.95 ± 1.15 | 11.50 ± 0.62 | 14.28 ± 4.42 |
| Ash (% dry weight) | 19.24 ± 1.08 | 18.77 ± 0.38 | 18.96 ± 4.81 |
| Energy (MJ kg-1 meat) | 20.75 ± 0.13 | 21.56 ± 0.14 | 20.63 ± 2.04 |
| **Condition indicators** | | | |
| C/N (atom C/ atom N) | 7.41 ± 0.10 | 6.50 ± 0.10 | 5.58 ± 0.07 |
| Protein/Energy (g protein/MJ) | 20.67 ± 0.07[a] | 23.20 ± 0.30[ab] | 25.63 ± 1.56[b] |
| Yield (% total wet weight) | 16.91 ± 0.33[a] | 23.70 ± 0.52[b] | 41.54 ± 0.68[c] |
| Wr (Relative weight index) | 1.17 ± 0.05[b] | 1.18 ± 0.04[b] | 1.02 ± 0.02[a] |

*chilensis* and the lowest by Chilean oysters (Table 1). Also, the lipid content showed significant differences in the mollusk tissue ($F_{2,6} = 7.50$; $P = 0.02$) (Table 1).

The C/N values did not present significant differences between the three bivalve mollusk species (mean = 6.5 ± 0.1). The relative weight index (Wr) showed a higher relative weight in oysters than mussels (13%). On the other hand, Chilean blue mussels registered the highest values in the protein/energy ratio ($H_{(2, n = 9)} = 6.54$; $P = 0.038$). The yield also showed differences between species, being significantly higher in *M. chilensis*, and lower in oysters (Table 1).

The fatty acid analyzes showed that the mussels had significantly lower SFA values (between 31% and 69% less) and PUFA (between 19% and 70% less) (Table 2). They also had lower values in the following acids: 15:0 (up to 93% less), 16:0 (up to 64% less), 18: 3n-3 (up to 91% less) and 22:6n-3 (up to 68% less). On the other hand, Chilean oysters presented the highest values in 14:0, 16:0, 18:0, 18:3n-3 and 20:4n-6 acids among the species analyzed (Table 2).

Principal component analyses explained 94.26% of all observed variance in the data. The first principal component (PC1) explained 78.12%, was negatively related to 16:0, 18:3n-3, 18:0, 14.0 and 20:4n-6 fatty acids. PC2 explained 16.14% of the variance of the data and was positively related to fatty acids 18:1n-9, 22:6n-3, 20:1n-11 and 20:5n-3 (Fig 2A). In graph PC1, the groups corresponding to the three species can be observed. Chilean blue mussels were

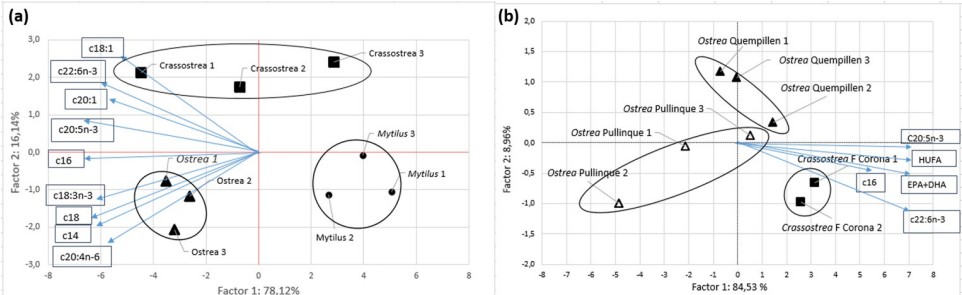

**Fig 2.** Plot of scores on principal components from the fatty acids: (a) Chilean oyster (*Ostrea chilensis*), Pacific oyster (*Crassostrea gigas*) and Chilean mussel (*Mytilus chilensis*) from Hueihue (Chiloé Island, Chile); (b) Chilean oyster (*Ostrea chilensis*) and Pacific oyster (*Crassostrea gigas*) from Faro Corona, Pullinque and Quempillén (Chiloé Island, Chile) (for locations see Fig 1). The ovals around the three separate clusters of points are drawn to aid interpretation; they do not represent confidence intervals.

**Table 2. Fatty acid content of oysters and mussels from Hueihue (Chiloé Island, Chile).** Each value represents the mean of three independent replicates ± standard error. The letters indicate significant differences between averages.

| | *Ostrea chilensis* | *Crassostrea gigas* | *Mytilus chilensis* |
|---|---|---|---|
| | Hueihue | Hueihue | Hueihue |
| **Fatty acids (mg fatty acid g$^{-1}$ dry weight of meat)** | | | |
| Myristic acid, 14:0 | 4.98 ± 0.37[b] | 2.25 ± 0.72[a] | 1.23 ± 0.30[a] |
| Pentadecanoic acid, 15:0 | 0.55 ± 0.04[b] | 0.35 ± 0.12[b] | 0.04 ± 0.04[a] |
| Palmitic acid, 16:0 | 16.08 ± 1.11[b] | 13.23 ± 3.67[ab] | 5.77 ± 1.36[a] |
| Heptadecanoic acid, 17:0 | 0.96 ± 0.07 | 0.65 ± 0.33 | 0.24 ± 0.04 |
| Stearic acid, 18:0 | 4.64 ± 0.23[b] | 2.44 ± 0.60[a] | 1.29 ± 0.27[a] |
| Oleic acid, 18:1n-9 | 1.91 ± 0.11 | 2.33 ± 0.61 | 0.40 ± 0.08 |
| Linoleic acid, 18:2n-6 | 1.28 ± 0.22 | 1.18 ± 0.29 | 0.42 ± 0.09 |
| Eicoosenoic acid, 20:1n-11 | 1.10 ± 0.08 | 1.13 ± 0.36 | 0.35 ± 0.08 |
| Linolenic acid, 18:3n-3 | 4.46 ± 0.39[c] | 2.00 ± 0.58[b] | 0.41 ± 0.09[a] |
| Eicosadienoic acid, 20:2n-6 | 0.22 ± 0.12 | 0.22 ± 0.09 | 0.18 ± 0.03 |
| Arachidonic acid, 20:4n-6 (ARA) | 1.99 ± 0.11[b] | 0.75 ± 0.25[a] | 0.57 ± 0.15[a] |
| Eicosapentaenoic acid, 20:5n-3 (EPA) | 15.14 ± 1.50 | 12.90 ± 3.64 | 5.30 ± 1.65 |
| Docosahexaenoic acid, 22:6n-3 (DHA) | 9.05 ± 1.08[b] | 8.89 ± 2.45[b] | 2.88 ± 0.85[a] |
| **Content of each group (mg g-1 meat)** | | | |
| SFA | 27.78 ± 1.66[b] | 19.29 ± 5.63[ab] | 8.70 ± 1.98[a] |
| MUFA | 5.01 ± 0.76 | 5.43 ± 1.76 | 3.49 ± 0.84 |
| PUFA | 32.31 ± 2.21[b] | 26.22 ± 7.46[ab] | 9.78 ± 2.79[a] |
| EPA + DHA | 24,19 ± 2.58 | 21,79 ± 6.09 | 8.18 ± 2.50 |
| Highly unsaturated fatty acids (HUFA) | 24.27 ± 2.53 | 21.83 ± 6.07 | 8.21 ± 2.52 |
| **Ratios** | | | |
| n-3/n-6 | 7.84 ± 0.62 | 10.14 ± 0.73 | 7.13 ± 1.82 |
| EPA/ARA | 7.62 ± 0.57[a] | 18.23 ± 1.88[b] | 9.22 ± 2.09[a] |
| DHA/EPA | 0.60 ± 0.01[a] | 0.69 ± 0.01[b] | 0.55 ± 0.01[a] |
| 16:1n-7/16:0 | 0.15 ± 0.04[a] | 0.12 ± 0.03[a] | 0.46 ± 0.02[b] |
| SFA/PUFA | 0.87 ± 0.09 | 0.73 ± 0.06 | 1.01 ± 0.27 |
| MUFA/PUFA | 0.15 ± 0.02 | 0.20 ± 0.02 | 0.42 ± 0.13 |
| HUFA/PUFA | 0.75 ± 0.02 | 0.83 ± 0.01 | 0.84 ± 0.04 |

separated from oysters due to their low C/N value and high protein/energy and yield values, while Pacific oysters were separated due to EPA/ARA and DHA/EPA values. The differentiating characteristics between the Chilean oyster and the other species were the high values of 18:3n-3, 20:4n-6, 20:5n-3 and 22:6n-3 (Fig 2A).

The EPA + DHA values were different between the three species. In this sense, the Chilean oyster was a better food to supply the daily requirement of EPA+DHA (250 mg EPA+DHA—suggested by Carboni et al. 2019). For 2018 samples, an amount of 174 g of Chilean mussel raw meat, or 65 g of Pacific oyster, or 59 g of Chilean oyster are needed to meet the recommended daily requirement (Fig 3).

## Analysis of oysters between locations

A higher protein content was estimated in Chilean flat oysters from Pullinque than in oysters from Quempillén. The Pacific oysters from Faro Corona had intermediate values (Table 3 - H$_{(2, n = 8)}$ = 5.55; $P = 0.06$). There were no significant differences in lipid and carbohydrate content between oyster species and localities (Table 3). The highest ash value was shown by the

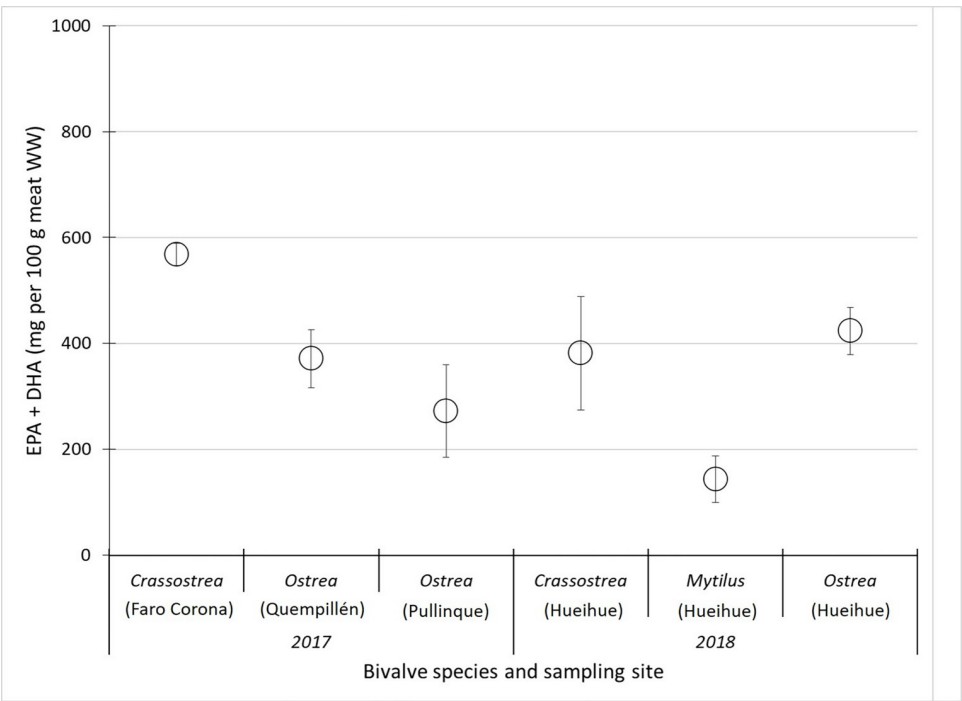

**Fig 3. Daily consumption of mg of EPA+DHA in 100 g of raw meat of *Crassostrea gigas, Ostrea chilensis* and *Mytilus chile*nsis from Faro Corona, Quempillén, Pullinque and Hueihue (Chiloé Island, Chile) harvested in winter.**

Pacific oyster and the lowest values were registered in the Chilean oysters ($F_{2, 5}$ = 13.60; $P$ = 0.009—Table 3).

The energy of *Ostrea chilensis* cultivated in Pullinque showed a significantly higher average value than the flat oysters from Quempillén ($F_{2,5}$ = 5.06; $P$ = 0.06). The protein/energy ratio was higher in the Chilean Pullinque oyster and in the Pacific oyster ($F_{2,5}$ = 21.02; $P$ = 0.004), while the C/N values did not present significant differences between the oysters and the sites (average = 6.3 ± 0.2—Table 3).

**Table 3. Proximal composition and condition index in oysters from Quempillén, Pullinque and Faro Corona (Chiloé Island, Chile).** Each value represents the average of three independent replicates ± standard error, except for *C. gigas* with two independent replicas ± standard error. The letters indicate significant differences between averages.

| | *O. chilensis* | *O. chilensis* | *C. gigas* |
|---|---|---|---|
| | (Quempillén) | (Pullinque) | (Faro Corona) |
| **Proximal composition** | | | |
| Protein (% dry weight) | 47.05 ± 0.47[a] | 53.39 ± 0.24[b] | 51.22 ± 1.89[ab] |
| Lipid (% dry weight) | 23.18 ± 0.30 | 22.36 ± 0.70 | 22.85 ± 0.65 |
| Carbohydrate (% dry weight) | 15.37 ± 1.20 | 11.65 ± 1.12 | 10.56 ± 2.57 |
| Ash (% dry weight) | 14.40 ± 0.46[b] | 12.60 ± 0.31[a] | 15.37 ± 0.04[b] |
| Energy (MJ kg$^{-1}$ meat) | 22.95 ± 0.05 | 23.49 ± 0.14 | 22.98 ± 0.26 |
| **Condition indicators** | | | |
| C/N (atom C/ atom N) | 6.48 ± 0.11 | 6.16 ± 0.15 | 6.26 ± 0.21 |
| Protein/Energy (g protein MJ$^{-1}$) | 20.50 ± 0.19[a] | 22.73 ± 0.13[b] | 22.29 ± 0.57[b] |
| Yield (% total wet weight) | 17.16 ± 0.34[b] | 15.07 ± 0.30[a] | 15.27 ± 0.36[a] |
| Wr (Relative weight index) | 0.93 ± 0.03[a] | 1.20 ± 0.04[b] | 0.88 ± 0.03[a] |

**Table 4. Fatty acid content in oyster meat from Quempillén, Pullinque and Faro corona (Chiloé Island, Chile).** Each value represents the average of independent replicates ± standard error, except for *C. gigas* with two independent replicas ± standard error. The letters show significant differences between averages.

| | *O. chilensis* | *O. chilensis* | *C. gigas* |
|---|---|---|---|
| | (Quempillén) | (Pullinque) | (Faro Corona) |
| **Fatty acids (mg fatty acid g$^{-1}$ dry weight of meat)** | | | |
| Myristic acid, 14:0 | 1.77 ± 0.15 | 2.14 ± 0.09 | 1.96 ± 0.22 |
| Pentadecanoic acid, 15:0 | 0.39 ± 0.03[a] | 0.51 ± 0.02[b] | 0.47 ± 0.01[ab] |
| Palmitic acid, 16:0 | 14.12 ± 0.70 | 11.56 ± 1.48 | 16.50 ± 0.84 |
| Heptadecanoic acid, 17:0 | 0.83 ± 0.09 | 0.95 ± 0.07 | 1.23 ± 0.06 |
| Stearic acid, 18:0 | 4.13 ± 0.44 | 3.39 ± 0.41 | 3.23 ± 0.34 |
| Oleic acid, 18:1n-9 | 3.11 ± 0.24 | 3.82 ± 1.07 | 3.74 ± 0.43 |
| Linoleic acid, 18:2n-6 | 1.81 ± 0.21 | 2.40 ± 1.27 | 1.47 ± 0.20 |
| Eicosenoic acid, 20:1n-9 | 2.99 ± 0.21[b] | 2.57 ± 0.53[b] | 1.62 ± 0.22[a] |
| Linolenic acid, 18:3n-3 | 3.58 ± 0.28 | 2.94 ± 0.33 | 2.76 ± 0.17 |
| Eicosadienoic acid, 20:2n-6 | 0.16 ± 0.10 | 0.54 ± 0.48 | n/a |
| Arachidonic acid, 20:4n-6 (ARA) | 1.62 ± 0.15 | 1.51 ± 0.08 | 1.21 ± 0.03 |
| Eicosapentaenoic acid, 20:5n-3 (EPA) | 12.54 ± 1.81 | 8.51 ± 2.97 | 18.24 ± 0.86 |
| Docosahexaenoic acid, 22:6n-3 (DHA) | 8.70 ± 1.32[ab] | 7.04 ± 2.01[a] | 14.22 ± 0.33[b] |
| Other saturated fatty acids (SFA) | 7.13 ± 0.73 | 6.19 ± 0.60 | 8.74 ± 0.34 |
| Other monounsaturated fatty acids (MUFA) | 0.64 ± 0.53 | 3.06 ± 2.63 | 1.32 ± 0.19 |
| Other polyunsaturated fatty acids (PUFA) | 3.89 ± 0.21 | 4.39 ± 0.50 | 3.07 ± 0.49 |
| **Total amount per group of fatty acids** | | | |
| SFA | 21.96 ± 1.17 | 17.53 ± 2.39 | 23.84 ± 1.29 |
| MUFA | 8.34 ± 0.40 | 8.93 ± 0.47 | 8.60 ± 0.60 |
| PUFA | 28.40 ± 3.49 | 22.52 ± 4.84 | 37.97 ± 0.88 |
| EPA + DHA | 21.24 ± 3.13 | 15.55 ± 4.98 | 32.46 ± 1.19 |
| Highly unsaturated fatty acids (HUFA) | 21.76 ± 2.89 | 15.63 ± 5.06 | 32.52 ± 1.24 |
| **Ratios** | | | |
| n-3/n-6 | 8.17 ± 0.39[ab] | 5.15 ± 1.77[a] | 12.86 ± 0.98[b] |
| EPA/ARA | 8.50 ± 0.49[a] | 7.23 ± 1.37[a] | 15.18 ± 1.07[b] |
| DHA/EPA | 0.69 ± 0.01 | 0.87 ± 0.07 | 0.78 ± 0.02 |
| 16:1n-7/16:0 | 0.15 ± 0.008 | 0.18 ± 0.03 | 0.15 ± 0.001 |
| SFA/PUFA | 0.79 ± 0.05 | 0.81 ± 0.09 | 0.63 ± 0.05 |
| MUFA/PUFA | 0.30 ± 0.03 | 0.44 ± 0.11 | 0.23 ± 0.02 |
| HUFA/PUFA | 0.76 ± 0.01 | 0.66 ± 0.09 | 0.86 ± 0.01 |

The fatty acid profile was similar in the two oyster species. However, there were significant differences in some fatty acids. For example, Chilean oysters had higher values of 20:1n-9 ($F_{2,5}$ = 8.20; $P$ = 0.03) and lower values of 22:6n-3 ($F_{2,5}$ = 4.62; $P$ = 0.07) than *Crassostrea gigas*. Also, the Chilean oysters from Pullinque showed higher values of 15:0 than the oysters from Quempillén ($F_{2,3}$ = 9.88; $P$ = 0.05—Table 4).

There were also significant differences in the relationships between fatty acids. In the n-3/n-6 relationship, the highest average value was recorded in Pacific oysters ($F_{2,5}$ = 8.26; $P$ = 0.03). The same occurred for EPA/ARA ($F_{2,3}$ = 16.79; $P$ = 0.02—Table 4).

The first two principal components explained 93.49% of the variability in the data. The PC1 and PC2 plots showed a slight separation between oyster species (Fig 2B). Chilean flat oysters from Pullinque and Quempillén formed two distinct but close groups. The Pullinque oysters occupied the lower left quadrant (with the exception of the point located in the upper right quadrant), and were characterized by lower values of EPA, DHA, 16:0, n-3/n-6 and yield.

On the other hand, the Pacific oyster formed a distinct group in the lower right quadrant (Fig 2B).

Principal component 1 (PC1) correlated positively with 20:5n-3, 16:0, 22:6n-3 fatty acid, HUFA and EPA+DHA (Fig 2B).

For 2017 samples, an amount of 67 g of Chilean oyster from Quempillén sampling site or 92 g of Chilean oyster from Pullinque or 44 g of Pacific oyster are needed to achieve daily requirement (Fig 3).

## Discussion

The concentration of omega-3 and omega-6 fatty acids are essential nutritional components for human health (i.e. prevent cardiovascular disease or depression–[37]). However, α-linolenic acid (18:3n-3, ALA) and linoleic acid (18:2n-6, LNA) cannot be synthesized within the human body, therefore they are essentials and must be obtained from the diet [38]. Besides, the main role of 18:3n-3 is to be precursor for biosynthesis of EPA and DHA, but its conversion to DHA is poor [39]. In this sense, WHO (World Health Organization) recommends including foods of marine origin because they have high concentrations of polyunsaturated fatty acids (PUFAs) [10], transforming them in functional foods, i.e., foods that provide beneficial physiological effects, not specifically nutritional, benefiting the health of consumers. Our results showed that both oysters and mussels can be considered a recommended option for a healthy human diet based on their high contribution of functional fatty acids n-3 LC-PUFA, specifically EPA and DHA [39]. The concentration of EPA ranged between 5.3 and 18.2 mg g$^{-1}$ dry weight, for DHA the range was 2.88–14.22 mg g$^{-1}$ dry weigh, with high n-3/n-6 ratios varying 5.2 and 12.9.

### Oysters vs mussel

Oysters and mussels are filter feeder bivalves with similar feeding behaviors. However, our results demonstrated that the acquisition and synthesis of fatty acids are species-specific. The Chilean oyster (*O. chilensis*), showed a high content of EPA and DHA above the values registered in the Chilean mussels (see Table 2). The fatty acid profile of oysters resembles an omnivorous feeder [40, 41].

The meat yield of the mussels was on average 56% higher than that of the oysters. Perhaps it is for this reason that the mussels' farming industry is successful in terms of meat yield. The meat yield values were similar to those reported previously by other authors (*Mytilus* sp–[42]; *Ostrea* sp–[43].

DHA/EPA values (<1) and 16:1n-7/16:0 (trophic marker) indicated that diatoms are an important food resource in all three species [43–45].

Stearic and oleic fatty acids are biomolecules of terrestrial origin; however, they have been found associated with aquaculture areas [46]. Stearic acid is normally found in particulate organic matter (POM) [44]. These particles come from terrigenous material washed away by the rains. In Chilean oysters we find the highest values of stearic acid. We can assume that this species of oyster is the one that consumes the most POM when compared to the other two bivalves.

Our results also showed that oleic, linoleic acids and DHA were more abundant in oysters than in mussels. These differences are likely to be associated with feeding strategies. Mussels ingest suspended particulate matter including inorganic material [47], while oysters have a diet that includes zooplankton, protozoa and/or dinoflagellates [44, 48, 49]. *Crassostrea gigas* and *Ostrea chilensis* had 8 and 14 times more pentadecanoic acid (15:0) than *M. chilensis*. The presence of this fatty acid is related to the intake of bacteria in the diet [49].

EPA+DHA values were different between the species studied. Presenting values higher than those reported for commercial species of bivalves from the northern hemisphere [50]. It has been reported that the recommended daily amount for human consumption should be 250 mg of EPA+DHA [51]. Therefore, 174 g of mussel meat (fresh), or 44 to 65 g of Pacific oyster, or 59 to 92 g of Chilean oyster should be consumed to meet the recommended daily requirement, considering the data obtained during the 2017 and 2018 sampling season. In this sense, the Chilean oyster is a good food to cover the daily requirement of EPA+DHA. Importantly, cooking techniques such as frying or boiling reduce the content of these essential fatty acids [52]. In addition, we found high values in the n-3/n-6 ratio for the oyster species analyzed (5.15 and 12.86). These values are much higher than the values reported for farmed and wild fish [53]. The higher this value, the healthier the diet, so oysters can be considered species of importance in human nutrition, opening a way to manage these nutrients through cultivation.

Western eating habits in recent times have drastically changed the nutritional content of food in developed countries, as a result of technological changes in the production and processing of high-calorie foods [54–56]. These have led to detrimental changes in nutrient metabolism leading to gene-diet interactions responsible for more obesity and systemic inflammation [57]. The role of n-3 PUFAs has been key in enhancing anti-inflammatory ability to control various chronic diseases, including atherosclerosis, coronary heart disease, diabetes, rheumatoid arthritis, depression, and cancer [58–60]. It has also been reported that n-3 and n-6 have a high potential for gastric cancer prevention and therapy [61, 62]. That is why the World Health Organization recommends including foods high in PUFA in daily consumption [63]. Under this context, the levels of n-3 LC-PUFA (mainly EPA and DHA) in oysters and mussels reported in this work may be beneficial for human health, and should be considered as a food alternative in the coming years.

## Differences between populations of oysters

The two species of oysters (*O. chilensis* y *C. gigas*) had a high content of proteins, lipids, and carbohydrates in winter 2017, which is positive for the aquaculture industry due to the commercial importance of these oyster species for the development of aquaculture on Chiloé Island. On the other hand, fatty acids showed significant differences between species, for example, eicosadienoic acid (c20: 2n-6) was detected in all species except for the Pacific oysters from Faro Corona. The Chilean oysters from Pullinque showed high values of pentadecanoic acid (15:0) when compared to the oysters from Quempillén. 15:0 is a naturally rare, straight chain saturated fatty acid [64] and it has been described as a fatty acid that may decrease mother-to-child transmission of HIV through breastfeeding [65]. This fatty acid has not been previously reported for oysters but has been reported for mussels [66]. We do not understand very well what these differences between the localities are due to but we do know that 15:0 has been found within the seston. Furthermore, 15:0 is considered a bacterial marker [49, 67] and has been related to the growth of bivalves (e.g., clams). The positive association of saturated fatty acids with growth may indicate utilization of carbon derived from bacteria in the mollusk diet [66]. Bacteria have been considered an important food in the diet of oysters, both in the larval stage and in adulthood. For example, they have been reported to increase larval survival of *Crassostrea gigas* [68].

This study showed that the environment plays an important role in the fatty acid acquisition of the Chilean oyster. The concentration of PUFAs in the oysters was different between the oysters of the analyzed localities (Quempillén, Pullinque and Hueihue), even though the localities were close. Therefore, nutrient acquisition is a quantitative variable that can be used to identify commercial species with higher fatty acid acquisition. In this way, growers could

have aquaculture productions with better nutritional characteristics [69]. However, it is necessary to understand the intra- and interspecific variability of the cultivated species, and our results contribute to the knowledge for these purposes.

One-hundred g of mussel or oyster meat provides a quarter of the daily amount of protein that an adult need. This same dose provides the recommended amount of vitamin $B_{12}$. In addition, these mollusks provide essential trace elements such as Se, Fe and Zn [14, 43, 70, 71]. However, the nutritional profiles of these bivalves will also change seasonally, so it is necessary to expand the study not only to more nutrients relevant to human health but also to all seasons of the year, especially during harvest periods. In conclusion, we consider that the nutritional variability of marine food sources should continue to be investigated, especially that providing high production volumes. Further investigations are needed on the heritability of the nutritional attributes of mussels and oysters to improve those characteristics in the short term.

## Conclusions

Shellfish sampled in winter were characterized by a high protein content, followed by medium values for lipid content and a low carbohydrate content compared to similar species in Europe. EPA + DHA values were different among the three species, with oysters showing the highest values in relation to mussels. It was also observed that the shellfish studied are rich in long-chain omega-3 polyunsaturated fatty acids (n-3 LC-PUFA), varying between 5.2–12.9 μg FA mg-1 dry weight with high n-3/n-6 ratios, depending on the species and the place of fattening. Keystone species can be considered as a sustainable option for a balanced and healthy human diet.

## Acknowledgments

To Justo García Campos from the town of Hueihue and Juan Pablo Fuentes Cid from Quempillén, for their collaboration in the installation of the experiments.

## Author Contributions

**Conceptualization:** Andrea Valenzuela, Ana Farias.

**Data curation:** Andrea Valenzuela, Ana Farias.

**Formal analysis:** Andrea Valenzuela.

**Funding acquisition:** Jorge E. Toro.

**Investigation:** Andrea Valenzuela, Pablo A. Oyarzún, Jorge E. Toro.

**Methodology:** Andrea Valenzuela.

**Project administration:** Andrea Valenzuela.

**Software:** Ana Farias.

**Supervision:** Andrea Valenzuela.

**Validation:** Jorge M. Navarro.

**Visualization:** Oscar Ramírez.

**Writing – original draft:** Andrea Valenzuela.

**Writing – review & editing:** Andrea Valenzuela, Pablo A. Oyarzún, Jorge E. Toro, Jorge M. Navarro, Ana Farias.

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
