## [Decision Letter · Decision Letter 0]

10 Dec 2021

PONE-D-21-28284Proximal and fatty acid analysis in Ostrea chilensis, Crassostrea gigas and Mytilus chilensis (Bivalvia: Mollusca) from Southern Chile.PLOS ONE

Dear Dr. Valenzuela,

Thank you for submitting your manuscript to PLOS ONE. After careful consideration, we feel that it has merit but does not fully meet PLOS ONE’s publication criteria as it currently stands. Therefore, we invite you to submit a revised version of the manuscript that addresses the points raised during the review process.

Both reviewers find merit in your study, but raise concerns. I agree that you should provide more details describing sampling conditions (abiotic factors, water depth) and reproductive status of the animals, as this has a strong impact on fatty acid profiles. Also, the method and results section needs to describe in more detail, how many replicate animals were used for each analysis and statistical tests. The manuscript also would benefit from being proof-read by a native speaker. 

We look forward to receiving your revised manuscript.

Kind regards,

Frank Melzner

Academic Editor

PLOS ONE

“Dr. Jorge Toro received the following funding:

Fondef ID19I10214 and ID16I10018”

“Fondef ID19I10214 and ID16I10018 financed this study. To Justo García Campos from the town of Hueihue and Juan Pablo Fuentes Cid from Quempillén, for their collaboration in the installation of the experiments.

“Dr. Jorge Toro received the following funding:

Fondef ID19I10214 and ID16I10018”

8. We note that Figure 1 in your submission contain [map/satellite] images which may be copyrighted. All PLOS content is published under the Creative Commons Attribution License (CC BY 4.0), which means that the manuscript, images, and Supporting Information files will be freely available online, and any third party is permitted to access, download, copy, distribute, and use these materials in any way, even commercially, with proper attribution. For these reasons, we cannot publish previously copyrighted maps or satellite images created using proprietary data, such as Google software (Google Maps, Street View, and Earth). For more information, see our copyright guidelines: http://journals.plos.org/plosone/s/licenses-and-copyright.

Reviewers' comments:

Reviewer's Responses to Questions

**Comments to the Author**

1. Is the manuscript technically sound, and do the data support the conclusions?

Reviewer #1: Partly

Reviewer #2: Yes

2. Has the statistical analysis been performed appropriately and rigorously? 

Reviewer #1: Yes

Reviewer #2: I Don't Know

3. Have the authors made all data underlying the findings in their manuscript fully available?

Reviewer #1: Yes

Reviewer #2: Yes

4. Is the manuscript presented in an intelligible fashion and written in standard English?

Reviewer #1: Yes

Reviewer #2: Yes

5. Review Comments to the Author

Reviewer #1: The paper by A. Valenzuela et al. describes interesting data on the proximal composition and fatty acid content of three mollusc species from Chile, that can contribute to the literature already existing on bivalve molluscs.

The Authors also tried to assess differences existing between species or even between the same species growing in relatively close locations.

On the other hand, several aspects of the paper need to be improved, especially in terms of description of experimental procedures, statistical analysis and results.

A list of suggestions/comments that could help the Authors during the revision stage is thus provided below, following the order of appearance in the manuscript.

Materials and methods

The title of the first sub-section (Sampling) of this section should be integrated to clarify that the sub-section deals also with the determination of proximal composition and fatty acid profile of molluscs.

Lines 122-129: the use of the same symbol, WW, first for the calculated wet weight (lines 125-126) and then for the experimental wet weight (lines 128-129) can be misleading. A different subscript (i.e., calc and exp, according to the case) should be added to the WW symbol to avoid any misunderstanding. Wr would thus be expressed as WWexp/WWcalc.

Line 133. Details should be provided about the use of the N x 6.25 rule. This can not immediately understandable for all readers.

Line 137 – 138. Which detector was used after GC separations? I guess that it was a FID but this detail has to be provided.

Lines 146-150. The sentence was already reported at lines 131-135.

Lines 151-152. Is there any previous reference about this approach to estimate the carbohydrate content?

Lines 157, 158, and 160-162. The “g dry weight” expression should be put between parentheses and then the -1 exponent should be placed outside; as an alternative, the word “per” could be placed before “g dry weight”.

Lines 173-176. The type of data used as original variables for PCA must be cited.

Results

Lines 188, 190, 193 and 196: the meaning of the H and F symbols needs to be clarified. Since the Authors claimed that normality and homoscedasticity tests were performed on data, the results of those tests should also be described.

Line 213. Fig2 should be referred to as Fig2a.

Figure 2. I would strongly suggest adding also loading plots to the figure, to help the readers in understanding how the variables influenced the differences observed between samples in the score plot. It is very complicated to switch to the Supplementary Material to find this information using Figure S1.

Tables 1-4. It seems that the results of tests for the comparison between means were not reported for all data, as significant differences could be supposed to be present sometimes (based on numerical values) but superscripts indicating the outcome of those tests were missing, thus suggesting that no significant difference was observed. Is this correct?

I would suggest using a consistent number of significant figures to express standard deviations (for example two) and then round off mean and standard deviation values accordingly.

The number of replicates should be clearly indicated in the tables captions.

Line 309. I would suggest using the word “quadrant” or “semi-plane”, according to the case, instead of “hemisphere”.

Lines 306-311: it would be better to use the term “principal” instead of “main”, since PCA results are described. I would be very careful in writing that Chilean oysters from Pullinque and Quempillen made up different groups in the score plot. Indeed, a remarkable variability was observed for the samples collected in Pullinque and at least one of them was very close to the samples from Quempillen.

Figure S1. What is the meaning of the colour code (and then of the legend) adopted in the figure?

Figure 3. The figure does not show daily consumptions, as stated in the caption, but the amounts of the two fatty acids referred to 100 g of meat of each mollusc. Please clarify this issue. The number of replicates used to calculate error bars should be indicated for each set of data.

Lines 375. Why were just those two values reported for the n-3/n-6 ratio? If the Authors meant to indicate the entire interval observed for those ratios in all the analyzed species, then the minimum value would be 5.15, found for Chilean oysters from Pullinque. A clarification is needed.

Lines 412-413. This sentence seems to be conceptually linked to the previous one, yet it concerns animal genetic improvement, so how can it be related to the sentence in which the focus was on the locality? Please clarify.

Conclusions. I would suggest rephrasing this section, focusing on more specific aspects of the work, like the comparison between oysters and mussels in terms of key components and the subtle effects that the location seemed to have on the nutrient content of the considered shellfish.

Reviewer #2: Present study determined the proximal biochemical composition and the fatty acid profile in the Chilean oyster (Ostrea chilensis), the Pacific oyster (Crassostrea gigas) and the Chilean mussel (Mytilus chilensis), to perform an intra and interspecific comparison. To my understanding the interspecific comparison (O. chilensis, C. gigas and M. chilensis) was performed on samples that were all collected from Hueihue in 2018, and the intraspecific comparison was meant on O. chilensis that were sampled from two locations (Quempillen vs. Pullinque) in 2017. Additionally to the results of O. chilensis (2017) the authors present results of C. gigas (sampled from Faro Corona in 2017). However, this is irritating as C. gigas is a different species and the results do not fit to the “intraspecific comparison”.

Besides minor grammatical inaccuracies, the ms is well written and the results seem to be reliable supporting the conclusion that the investigated shellfish species can be a “sustainable option for a balanced and healthy human diet”. However, the finding that oysters and mussels are healthy food components in human diet is not new and the study will benefit from a clearer (hypothesis-driven) research question to more strongly emphasize its relevance to the scientific community. In addition, the ms can be improved by a more detailed description of the M&M part and an in-depth discussion.

Major comments:

1) The authors aim to “nutritionally characterize the food quality” of two oyster and one mussel species from aquaculture in Southern Chile. To improve the study, the results presented (content of lipids, protein etc. as well as fatty acid composition) need to be put in context with literature data and discussed appropriately; e.g. Do the results differ from bivalves obtained from European or US aquaculture facilities? If so, why? Investigations were conducted on bivalves collected during the winter season, and the data could be discussed/compared with data from bivalves collected during spring/summer.

2) Please highlight why the different locations were chosen for the intraspecific comparison. In what respect do they differ from each other? The intraspecific comparison aspect should be more clearly elaborated in the discussion. In its present form (under the heading “Differences between populations of oysters”) it is a bit confusing as the dataset of C. gigas is also discussed. Please comment on your decision to add the data of C. gigas (obtained in 2017) to the results part of the intraspecific comparison (line 243f).

3) With respect to the results in Table 1 showing highest proximal composition values in protein content (42 – 52% dry weight) and lower values in lipid (14.5 – 19.5%) and carbohydrates (11.5 to 20.9%), the statement “shellfish sampled in winter were characterized by high lipid content along with low protein and carbohydrates contents (line 36f, and 431f) is irritating. Please clarify.

Specific comments:

Line 26/27: Please clarify the definitions of equal contribution of the authors - ¶ vs. &.

Line 83: Change into “On the one hand…”

Line 107f: Sampling of animals: Please clarify the sentence "3 samples of 20 oysters" as it is 60 animals in total, isn't it? Please add more details about the environmental parameters at the time of collection (water temperature, salinity...) and transport of the animals (duration, transport on ice/water...). Were the animals kept in institutional aquarium systems prior to dissection? If yes, please add details on husbandry.

Line 121ff: Please comment on using the calculation of yield instead of condition index (soft body dry mass*100/shell dry mass) which is more common. In any case, data sets should be compared with literature data to increase the reliability of the results.

Line 122: Please add the underlying data for the calculation of the allometric relationship of WW. The cited reference (#30) is for Wr for fish. Is it also used for shellfish?

Line 133: Please clarify how samples were treated prior to analysis to ensure homogeneity, as parameters were determined in whole animal extracts.

Line 133: Unclear, please add more information how crude protein was calculated.

Line 146f: Repetition, see line 131f.

Line 161: Please provide more details concerning the equation parameters “12g” and “15g”.

Line 163: Consider rephrasing “performance”. If it is a common term in aquaculture, please provide a citation for the calculation.

Line 165f: Statistics: Missing info of p level.

Line 191: Revise sentence “higher compared to …”. Please check the whole ms regarding this aspect. Furthermore, using the mean values of lipid content (Table 1) the values (9 to 25%) do not fit. Please clarify.

Line 202: Table 1 and all other Tables: Please add experimental number. Energy is expressed as MJ per kg dry meat, isn’t it? Please specify that “different” letters indicate differences.

Line 206: see comments to line 191.

Line 317f: What about the EDA+DHA concentration of nearly 600mg/100g ww meat determined in C. gigas samples collected in 2017?

Line 368f: This is a repetition, see line 317f.

Line 375: Please clarify: 7.13 “y” 12.86.

Line 376: Please add more details/information. The reader wonders, so what?

6. PLOS authors have the option to publish the peer review history of their article (what does this mean?). If published, this will include your full peer review and any attached files.

Reviewer #1: No

Reviewer #2: No

---

## [Author Response · Author response to Decision Letter 0]

20 Jan 2022

REPLIES TO REVIEWERS

PONE-D-21-28284

Proximal and fatty acid analysis in Ostrea chilensis, Crassostrea gigas and Mytilus chilensis (Bivalvia: Mollusca) from Southern Chile.

Dear Academic Editor

Below you can find the corrections made in the manuscript according to the suggestions made by you and reviewers #1 and 2:

ACADEMIC EDITOR

A.E: Both reviewers find merit in your study, but raise concerns. I agree that you should provide more details describing sampling conditions (abiotic factors, water depth) and reproductive status of the animals, as this has a strong impact on fatty acid profiles. Also, the method and results section need to describe in more detail, how many replicate animals were used for each analysis and statistical tests. The manuscript also would benefit from being proof-read by a native speaker. 

Reply: we took these considerations into account and they were included in the manuscript in each of the sections.

A.E: Please include the following items when submitting your revised manuscript:

Reply: The three files mentioned in this paragraph were created and submitted separately.

A.E: Reply: Without changes

JOURNAL REQUIREMENTS:

1. When submitting your revision, we need you to address these additional requirements. Please ensure that your manuscript meets PLOS ONE's style requirements, including those for file naming. The PLOS ONE style templates can be found at https://journals.plos.org/plosone/s/file?id=wjVg/PLOSOne_formatting_sample_main_body.pdf and https://journals.plos.org/plosone/s/file?id=ba62/PLOSOne_formatting_sample_title_authors_affiliations.pdf

Reply: all the requirements requested by PLOS ONE were considered.

Reply: Additional permits are not required for field work. All locations are freely accessible and the collection of samples for research is allowed.

Reply: the manuscript was reviewed by a native speaker (Michael Note) before being resubmitted.

 4. Thank you for stating the following financial disclosure: “Dr. Jorge Toro received the following funding: Fondef ID19I10214 and ID16I10018”

Reply: Dr. Jorge Toro worked on the design of the study, the decision to publish, and the preparation of the manuscript. An amended statement of the funder's role will be included in the cover letter. 

“Fondef ID19I10214 and ID16I10018 financed this study. To Justo García Campos from the town of Hueihue and Juan Pablo Fuentes Cid from Quempillén, for their collaboration in the installation of the experiments.

“Dr. Jorge Toro received the following funding:

Fondef ID19I10214 and ID16I10018”

Reply: the codes of the projects responsible for the financing were removed from the acknowledgments section. Codes were placed on the application and an amended statement of project funding was included in the cover letter.

Reply: The data from this research can be found in the tables and figures presented in the manuscript and in the supplementary material, and are freely available to the journal and those who work in bivalve mollusc nutrition research.

Reply: the ethics statement was included in the methods section of the manuscript, including the full name of the ethics committee that approved the study.

 8. We note that Figure 1 in your submission contain [map/satellite] images which may be copyrighted. All PLOS content is published under the Creative Commons Attribution License (CC BY 4.0), which means that the manuscript, images, and Supporting Information files will be freely available online, and any third party is permitted to access, download, copy, distribute, and use these materials in any way, even commercially, with proper attribution. For these reasons, we cannot publish previously copyrighted maps or satellite images created using proprietary data, such as Google software (Google Maps, Street View, and Earth). For more information, see our copyright guidelines: http://journals.plos.org/plosone/s/licenses-and-copyright.

Respuesta: Figure 1 of the manuscript corresponds to a map made by the main author of this manuscript, using the ArcGIS Program.

REVIEWERS' COMMENTS:

Reviewer's Responses to Questions

Comments to the Author

1. Is the manuscript technically sound, and do the data support the conclusions?

Reviewer #1: Partly

Reviewer #2: Yes

REPLY: the manuscript was made based on a robust and solid research work, with data that support the conclusions. All considerations were taken to carry out a rigorous experimental design, with adequate controls, replicates and sample sizes.

2. Has the statistical analysis been performed appropriately and rigorously?

Reviewer #1: Yes

Reviewer #2: I Don't Know

REPLY: The treatment of the data was carried out in an adequate and rigorous manner according to the nature of the data.

3. Have the authors made all data underlying the findings in their manuscript fully available?

Reviewer #1: Yes

Reviewer #2: Yes

REPLY: The data is available to the journal and for universal use by those working in science, especially in marine bivalve nutrition.

4. Is the manuscript presented in an intelligible fashion and written in standard English?

Reviewer #1: Yes

Reviewer #2: Yes

REPLY: the manuscript is written in standard English, proofread and proofread by native speaker Michael Note.

5. Review Comments to the Author

Reviewer #1: The paper by A. Valenzuela et al. describes interesting data on the proximal composition and fatty acid content of three mollusc species from Chile, that can contribute to the literature already existing on bivalve molluscs.

The Authors also tried to assess differences existing between species or even between the same species growing in relatively close locations.

On the other hand, several aspects of the paper need to be improved, especially in terms of description of experimental procedures, statistical analysis and results.

A list of suggestions/comments that could help the Authors during the revision stage is thus provided below, following the order of appearance in the manuscript.

Materials and methods

The title of the first sub-section (Sampling) of this section should be integrated to clarify that the sub-section deals also with the determination of proximal composition and fatty acid profile of molluscs.

Reply: Done

Lines 122-129: the use of the same symbol, WW, first for the calculated wet weight (lines 125-126) and then for the experimental wet weight (lines 128-129) can be misleading. A different subscript (i.e., calc and exp, according to the case) should be added to the WW symbol to avoid any misunderstanding. Wr would thus be expressed as WWexp/WWcalc.

Reply: The WW values cited between lines 122-129 are any WW obtained experimentally to apply the index Wr, that allows to determine if a WW sampled in an individual of species A relates to the expected WLR for species A with a values inferior, similar or superior to 1; ie, there are no calculated WW values, only experimental values in all cases

Line 133. Details should be provided about the use of the N x 6.25 rule. This can not immediately understandable for all readers.

Reply: In nutrition, the total nitrogen content of the sample is converted into a value called "crude protein" by multiplying it by 6.25, a constant based on the conventional assumption that any protein is composed of 16% nitrogen.

Line 137 – 138. Which detector was used after GC separations? I guess that it was a FID but this detail has to be provided.

Reply: Flame ionization detector (FID) Thermo Scientific was used

Lines 146-150. The sentence was already reported at lines 131-135.

Reply: corrected

Lines 151-152. Is there any previous reference about this approach to estimate the carbohydrate content?

Reply: the methods of substraction of total carbohydrate from proximal composition as cited by Metzger LE and Nielsen SS, 2019. Chapter 3 Nutrition Labelling. In: Food Analysis. Fifth edition, SS Nielsen (ed), Food Science Text Series, Springer, Cham, Switzerland, pp 35-44

Lines 157, 158, and 160-162. The “g dry weight” expression should be put between parentheses and then the -1 exponent should be placed outside; as an alternative, the word “per” could be placed before “g dry weight”.

Reply: done

Lines 173-176. The type of data used as original variables for PCA must be cited.

Reply: The data used for PCA analysis were the values of each replicate for: every fatty acid, every relationship between fatty acids, every performance indicator, and every proximal content. Finally, only those variables most correlated with the first two components were selected for final PCA analysis.

Results

Lines 188, 190, 193 and 196: the meaning of the H and F symbols needs to be clarified. Since the Authors claimed that normality and homoscedasticity tests were performed om data, the results of those tests should also be described.

Reply: done

Line 213. Fig2 should be referred to as Fig2a.

Reply: done

Figure 2. I would strongly suggest adding also loading plots to the figure, to help the readers in understanding how the variables influenced the differences observed between samples in the score plot. It is very complicated to switch to the Supplementary Material to find this information using Figure S1.

Reply: Figure 2a and b was improved and replaced with data from the supplementary material. Therefore, the supplementary material was eliminated, and its data were incorporated along with the variables in Figure 2, for a better understanding of the results.

Tables 1-4. It seems that the results of tests for the comparison between means were not reported for all data, as significant differences could be supposed to be present sometimes (based on numerical values) but superscripts indicating the outcome of those tests were missing, thus suggesting that no significant difference was observed. Is this correct?

I would suggest using a consistent number of significant figures to express standard deviations (for example two) and then round off mean and standard deviation values accordingly.

The number of replicates should be clearly indicated in the tables captions.

Reply: done.

Line 309. I would suggest using the word “quadrant” or “semi-plane”, according to the case, instead of “hemisphere”.

Reply: done.

Lines 306-311: it would be better to use the term “principal” instead of “main”, since PCA results are described. I would be very careful in writing that Chilean oysters from Pullinque and Quempillen made up different groups in the score plot. Indeed, a remarkable variability was observed for the samples collected in Pullinque and at least one of them was very close to the samples from Quempillen.

Reply: done.

Figure S1. What is the meaning of the colour code (and then of the legend) adopted in the figure?

Reply: In blue (active) it refers to the active variables that influence the principal components and in red (suppl) to the supplementary variables that have no influence on the analysis of the principal components, but do help to interpret the results. However, this graph was corrected and the supplementary material was removed, Figure 2 and Figure S1 were merged for a better understanding of the results.

Figure 3. The figure does not show daily consumptions, as stated in the caption, but the amounts of the two fatty acids referred to 100 g of meat of each mollusc. Please clarify this issue. The number of replicates used to calculate error bars should be indicated for each set of data.

Reply: Figure 3. Potential intake of EPA + DHA in a 100 g edible portion from Crassostrea gigas, Ostrea chilensis and Mytilus chilensis.

Lines 375. Why were just those two values reported for the n-3/n-6 ratio? If the Authors meant to indicate the entire interval observed for those ratios in all the analyzed species, then the minimum value would be 5.15, found for Chilean oysters from Pullinque. A clarification is needed.

Reply: corrected

Lines 412-413. This sentence seems to be conceptually linked to the previous one, yet it concerns animal genetic improvement, so how can it be related to the sentence in which the focus was on the locality? Please clarify.

Reply: corrected

Conclusions. I would suggest rephrasing this section, focusing on more specific aspects of the work, like the comparison between oysters and mussels in terms of key components and the subtle effects that the location seemed to have on the nutrient content of the considered shellfish.

Reply: Reworded this section as suggested by reviewer.

Reviewer #2: Present study determined the proximal biochemical composition and the fatty acid profile in the Chilean oyster (Ostrea chilensis), the Pacific oyster (Crassostrea gigas) and the Chilean mussel (Mytilus chilensis), to perform an intra and interspecific comparison. To my understanding the interspecific comparison (O. chilensis, C. gigas and M. chilensis) was performed on samples that were all collected from Hueihue in 2018, and the intraspecific comparison was meant on O. chilensis that were sampled from two locations (Quempillen vs. Pullinque) in 2017. Additionally, to the results of O. chilensis (2017) the authors present results of C. gigas (sampled from Faro Corona in 2017). However, this is irritating as C. gigas is a different species and the results do not fit to the “intraspecific comparison”.

Besides minor grammatical inaccuracies, the ms is well written and the results seem to be reliable supporting the conclusion that the investigated shellfish species can be a “sustainable option for a balanced and healthy human diet”. However, the finding that oysters and mussels are healthy food components in human diet is not new and the study will benefit from a clearer (hypothesis-driven) research question to more strongly emphasize its relevance to the scientific community. In addition, the ms can be improved by a more detailed description of the M&M part and an in-depth discussion.

Major comments:

1) The authors aim to “nutritionally characterize the food quality” of two oyster and one mussel species from aquaculture in Southern Chile. To improve the study, the results presented (content of lipids, protein etc. as well as fatty acid composition) need to be put in context with literature data and discussed appropriately; e.g. Do the results differ from bivalves obtained from European or US aquaculture facilities? If so, why? Investigations were conducted on bivalves collected during the winter season, and the data could be discussed/compared with data from bivalves collected during spring/summer.

Reply: This comment has been considered in the discussion to deepen the comparison between the three species of bivalves.

2) Please highlight why the different locations were chosen for the intraspecific comparison. In what respect do they differ from each other? The intraspecific comparison aspect should be more clearly elaborated in the discussion. In its present form (under the heading “Differences between populations of oysters”) it is a bit confusing as the dataset of C. gigas is also discussed. Please comment on your decision to add the data of C. gigas (obtained in 2017) to the results part of the intraspecific comparison (line 243f).

Reply: The locations chosen in this study for the comparison between oyster species were selected because Quempillén and Pullinque correspond to two areas of natural banks of O. chilensis on Chiloé Island. The town of Quempillén is an estuarine zone with high salinity fluctuations. Instead, the town of Pullinque is a bay in the Gulf of Quetalmahue with depths of 7 to 8 meters. The species C. gigas was added because it is the second most important commercially important oyster species in Chile and because it also has natural banks on Chiloé Island.

3) With respect to the results in Table 1 showing highest proximal composition values in protein content (42 – 52% dry weight) and lower values in lipid (14.5 – 19.5%) and carbohydrates (11.5 to 20.9%), the statement “shellfish sampled in winter were characterized by high lipid content along with low protein and carbohydrates contents (line 36f, and 431f) is irritating. Please clarify.

Reply: The paragraphs mentioned in lines 36f and 431f were corrected.

Specific comments:

Line 26/27: Please clarify the definitions of equal contribution of the authors - ¶ vs. &.

Reply: the authors' equal contribution to:

 ¶ = all of these authors contributed to study design, data collection and analysis, decision to publish, or manuscript preparation.

&= all these authors contributed to the experimental and field work.

Line 83: Change into “On the one hand…”

Reply: done.

Line 107f: Sampling of animals: Please clarify the sentence "3 samples of 20 oysters" as it is 60 animals in total, isn't it? Please add more details about the environmental parameters at the time of collection (water temperature, salinity...) and transport of the animals (duration, transport on ice/water...). Were the animals kept in institutional aquarium systems prior to dissection? If yes, please add details on husbandry.

Reply: For each locality, 60 sampled specimens were collected from cultivation areas, where 3 samples (replicates) of 20 individuals were randomly made, except for C. gigas, which were fewer specimens (40 individuals). Then, in the laboratory, the tissues without the shell were worked, each sample was stored in a refrigerated/frozen ziplox bag with the meat of 20 individuals corresponding to replica 1, 2 and 3 of each sampling.

Line 121ff: Please comment on using the calculation of yield instead of condition index (soft body dry mass*100/shell dry mass) which is more common. In any case, data sets should be compared with literature data to increase the reliability of the results.

Reply: The samples were processed in the field, where the wet weight was directly measured. In our laboratory, the samples were lyophilized to be considered for proximal content and fatty acid content. The work of Sing & Ransangan (2019) measure yield according to wet weight.

Sing, O. F., & Ransangan, J. (2019). Effect of physicochemical parameters and phytoplankton composition on growth performance of green mussel (Perna viridis) in Ambong Bay and Marudu Bay, Sabah, Malaysia. Journal of Fisheries and Environment, 43(1), 50-68.

Line 122: Please add the underlying data for the calculation of the allometric relationship of WW. The cited reference (#30) is for Wr for fish. Is it also used for shellfish?

Reply: Now the underlying data suggested is: 

The relative weight index [30] is based on the length -weight relationships (LWRs). The total wet weight in grams (WW), and shell length in milimeters (L) obtained from each individual sampled during the study were used. The wet weights were fit using a power regression as a function of shell lenght to obtain WW = aLb , where a is the intercept (condition factor) and b is the slope (relative growth) following Sousa et al (2020 a, b). The relationships were WW = 0.021*L2.16 for M. chilensis with 60 individuals (R2= 0.48), WW = 0.0004*L2.77 for O. chilensis with 180 individuals (R2= 0.51), and WW = 0.079*L1.48 for C. gigas with 100 individuals (R2= 0.67).

The use of WLRs are commonly used for shellfish see Sousa et al 2020a, b. The WLRs are used to compare different ecological, geographical, reproductive, etc., conditions for some particular species, and the comparisons is about the exponent of WLRs. 

So the Wr is only a ratio WW/WLR to show if the animals are more slim or fat respect to an average condition represented by WLR of the species in different geographical areas, or growth conditions, or etc, Therefore it is applicable to shellfish, the same as WLRs

Line 133: Please clarify how samples were treated prior to analysis to ensure homogeneity, as parameters were determined in whole animal extracts.

Reply: For each sample of 20 individuals the tissues were pooled and freeze dried in a Savant freeze dryer (-80°C), then they were ground to homogenize and saved at -41°C until the respective biochemical analysis.

Line 133: Unclear, please add more information how crude protein was calculated.

Reply: In nutrition, the total nitrogen content of the sample is converted into a value called "crude protein" by multiplying it by 6.25, a constant based on the conventional assumption that any protein is composed of 16% nitrogen.

Line 146f: Repetition, see line 131f.

Reply: corrected

Line 161: Please provide more details concerning the equation parameters “12g” and “15g”.

Reply: correspond to the molecular weights of carbon and nitrogen. The nitrogen value was corrected to 14.

Line 163: Consider rephrasing “performance”. If it is a common term in aquaculture, please provide a citation for the calculation.

Reply: CI commercial = wet meat weight x 100 /whole wet weight (live) = Meat wet weight x 100/ Total wet weight following Hickman and Illingworth (1980)

Line 165f: Statistics: Missing info of p level.

Reply: p< 0.05

Line 191: Revise sentence “higher compared to …”. Please check the whole ms regarding this aspect. Furthermore, using the mean values of lipid content (Table 1) the values (9 to 25%) do not fit. Please clarify.

Reply: corrected

Line 202: Table 1 and all other Tables: Please add experimental number. Energy is expressed as MJ per kg dry meat, isn’t it? Please specify that “different” letters indicate differences.

Reply: done

Line 206: see comments to line 191.

Reply: done

Line 317f: What about the EDA+DHA concentration of nearly 600mg/100g ww meat determined in C. gigas samples collected in 2017?

Reply: For 2017 samples, an amount of 67 g of Chilean oyster from Quempillén sampling site or 92 g of Chilean oyster from Pullinque or 44 g of Pacific oyster are needed to achieve daily requirement (Fig.3).

Line 368f: This is a repetition, see line 317f.

Reply: corrected

Line 375: Please clarify: 7.13 “y” 12.86.

Reply: corrected.

Line 376: Please add more details/information. The reader wonders, so what?

Reply: Reworded this paragraph to be more consistent.

---

## [Decision Letter · Decision Letter 1]

5 Apr 2022

PONE-D-21-28284R1Proximal and fatty acid analysis in Ostrea chilensis, Crassostrea gigas and Mytilus chilensis (Bivalvia: Mollusca) from Southern Chile.PLOS ONE

Dear Dr. Valenzuela,

Thank you for submitting your manuscript to PLOS ONE. After careful consideration, we feel that it has merit but does not fully meet PLOS ONE’s publication criteria as it currently stands. Therefore, we invite you to submit a revised version of the manuscript that addresses the points raised during the review process.

Please ensure all comments made by reviewer 2 are fully addressed and that important information is added to the manuscript. 

We look forward to receiving your revised manuscript.

Kind regards,

Frank Melzner

Academic Editor

PLOS ONE

Reviewers' comments:

Reviewer's Responses to Questions

**Comments to the Author**

1. If the authors have adequately addressed your comments raised in a previous round of review and you feel that this manuscript is now acceptable for publication, you may indicate that here to bypass the “Comments to the Author” section, enter your conflict of interest statement in the “Confidential to Editor” section, and submit your "Accept" recommendation.

Reviewer #1: All comments have been addressed

Reviewer #2: (No Response)

2. Is the manuscript technically sound, and do the data support the conclusions?

Reviewer #1: (No Response)

Reviewer #2: Partly

3. Has the statistical analysis been performed appropriately and rigorously? 

Reviewer #1: (No Response)

Reviewer #2: N/A

4. Have the authors made all data underlying the findings in their manuscript fully available?

Reviewer #1: (No Response)

Reviewer #2: Yes

5. Is the manuscript presented in an intelligible fashion and written in standard English?

Reviewer #1: (No Response)

Reviewer #2: Yes

6. Review Comments to the Author

Reviewer #1: (No Response)

Reviewer #2: xxxxxxxxxxxxxxxxxxxxxxxxxxxxxxxxxxxxxxxxxxxxxxxxxxxxxxxxxxxxxxxxxxxxxxxxxxxxxxxxxxxxxxxxxxxxxxxxxxxxxxxxxxxxxxxxxxxxxxxxxxxxxxxxxxxxxxxxxxxxxxxxxxxxxxxxxxxxxxxxxxxxxxxxxxxxxxxxxxxxxxxx

7. PLOS authors have the option to publish the peer review history of their article (what does this mean?). If published, this will include your full peer review and any attached files.

Reviewer #1: No

Reviewer #2: No

---

## [Author Response · Author response to Decision Letter 1]

28 Apr 2022

REPLIES TO REVIEWER

PONE-D-21-28284

Proximal and fatty acid analysis in Ostrea chilensis, Crassostrea gigas and Mytilus chilensis (Bivalvia: Mollusca) from Southern Chile.

Dear Academic Editor

Below you can find the corrections made in the manuscript according to the suggestions made by the reviewer # 2:

REVIEWER #2 COMMENTS:

Line 26/27: Please clarify the definitions of equal contribution of the authors - ¶ vs. &.

Reply: the authors' equal contribution to:

 ¶ = all of these authors contributed to study design, data collection and analysis, decision to publish, or manuscript preparation.

&= all these authors contributed to the experimental and field work.

Thank you, but this information should also be in the paper, shouldn't it?

Reply: We appreciate your clarification. Your suggestion will be included in the manuscript.

Line 83: Change into “On the one hand…”

Reply: done.

hm,, the authors revised the sentence but not as suggested. That's fine but the reply "done" is irritating. 

Reply: We regret the incorrect use of the word "done". We will take this into account in future corrections.

Line 107f: Sampling of animals: Please clarify the sentence "3 samples of 20 oysters" as it is 60 animals in total, isn't it? Please add more details about the environmental parameters at the time of collection (water temperature, salinity...) and transport of the animals (duration, transport on ice/water...). Were the animals kept in institutional aquarium systems prior to dissection? If yes, please add details on husbandry.

Reply: For each locality, 60 sampled specimens were collected from cultivation areas, where 3 samples (replicates) of 20 individuals were randomly made, except for C. gigas, which were fewer specimens (40 individuals). Then, in the laboratory, the tissues without the shell were worked, each sample was stored in a refrigerated/frozen ziplox bag with the meat of 20 individuals corresponding to replica 1, 2 and 3 of each sampling.

Please add more details….” This comment wasn't addressed, neither in the reply nor in the revised ms. 

Reply: We improved the paragraph, including the required information in the manuscript. We appreciate your helpful suggestions.

Line 121ff: Please comment on using the calculation of yield instead of condition index (soft body dry mass*100/shell dry mass) which is more common. In any case, data sets should be compared with literature data to increase the reliability of the results.

Reply: Samples were processed in the field, where 20 individuals were randomly selected for each sample, measured and weighed wet, shells were immediately removed and the soft bodies of the 20 individuals were grouped in a properly labeled bag. The bags were stored cold for transport to the laboratory. In the laboratory, the samples were frozen, freeze-dried, macerated and grouped by bag, kept frozen until analysis for proximal and fatty acid content.

The index was calculated according to Hickmann and Illingworth (1980).

These authors compared different condition index formulas that are weight/weight ratios and that reflect changes in body proportions in bivalve populations.

CIwet = [wet meat volumen/(whole volumen - Shell volumen)] * 100

CIvolume = [dry meat wt / (whole volumen - shell volumen)] * 100

CIweight = [dry meat wt/(whole wt - shell wt)] * 100

CIcommercial = [Meat wet wt / Total wet weight] * 100

They found that CIcommercial correlated 92.7% with CIwet, 90.4% with CIvolume and 90.9% with CIweight.

According to Hickmann and Illingworth (1980), although CIweight is the most advisable for reliable comparisons, it is possible to standardize the measurement of wet weight in an index such as CIcommercial to minimize errors by always adopting the same weighing routines, which is what was applied in this study.

Line 133: Please clarify how samples were treated prior to analysis to ensure homogeneity, as parameters were determined in whole animal extracts.

Reply: For each sample of 20 individuals the tissues were pooled and freeze dried in a Savant freeze dryer (-80°C), then they were ground to homogenize and stored at -41°C until the respective biochemical analysis.

This is an important information that the tissue of 20 individuals was pooled prior to analyses and must be given in the ms. 

Reply: Thank you for the clarification. This information has been included in the manuscript.

Line 133: Unclear, please add more information how crude protein was calculated.

Reply: In nutrition, the total nitrogen content of the sample was converted into a value called "crude protein" by multiplying it by 6.25, a constant based on the conventional assumption that any protein is composed of 16% nitrogen.

Again, this is an information that should be given in the ms. 

Reply: Thank you for the clarification. This information has also been included in the manuscript.

Line 146f: Repetition, see line 131f.

Reply: corrected

Sorry, the repetition is still in the revised version. 

Reply: This repetition has been removed. Thanks for the clarification

---

## [Editor Report · Decision Letter 2]

30 May 2022

PONE-D-21-28284R2Proximal and fatty acid analysis in Ostrea chilensis, Crassostrea gigas and Mytilus chilensis (Bivalvia: Mollusca) from Southern Chile.PLOS ONE

Dear Dr. Valenzuela,

Thank you for submitting your manuscript to PLOS ONE. After careful consideration, we feel that it has merit but does not fully meet PLOS ONE’s publication criteria as it currently stands. Therefore, we invite you to submit a revised version of the manuscript that addresses the points raised during the review process. Please make sure to add a complete point-by-point reply to comments raised by reviewer 2 that fully addresses the comments and make sure to incorporate vital information in the ms and not just in the response to the reviewer.

We look forward to receiving your revised manuscript.

Kind regards,

Frank Melzner

Academic Editor

PLOS ONE
---

## [Author Response · Author response to Decision Letter 2]

2 Jun 2022

Dear Academic Editor

The points suggested by reviewer 2 in the second revision and also the rest of the original points raised by reviewer 1 and 2 in the first revision were fully addressed.

I appreciate your support and understanding.

Below you can find the corrections made in the manuscript according to the suggestions made in the first (reviewer 1 and 2) and second revision (reviewer 2).

REPLIES TO REVIEWER

PONE-D-21-28284

Proximal and fatty acid analysis in Ostrea chilensis, Crassostrea gigas and Mytilus chilensis (Bivalvia: Mollusca) from Southern Chile.

Second revision R2 (May 2022)

REVIEWER #2 COMMENTS:

Line 26/27: Please clarify the definitions of equal contribution of the authors - ¶ vs. &.

Reply: the authors' equal contribution to:

 ¶ = all of these authors contributed to study design, data collection and analysis, decision to publish, or manuscript preparation.

&= all these authors contributed to the experimental and field work.

Thank you, but this information should also be in the paper, shouldn't it?

Reply: We appreciate your clarification. Your suggestion will be included in the manuscript.

Line 83: Change into “On the one hand…”

Reply: done.

hm,, the authors revised the sentence but not as suggested. That's fine but the reply "done" is irritating. 

Reply: We regret the incorrect use of the word "done". We will take this into account in future corrections.

Line 107f: Sampling of animals: Please clarify the sentence "3 samples of 20 oysters" as it is 60 animals in total, isn't it? Please add more details about the environmental parameters at the time of collection (water temperature, salinity...) and transport of the animals (duration, transport on ice/water...). Were the animals kept in institutional aquarium systems prior to dissection? If yes, please add details on husbandry.

Reply: For each locality, 60 sampled specimens were collected from cultivation areas, where 3 samples (replicates) of 20 individuals were randomly made, except for C. gigas, which were fewer specimens (40 individuals). Then, in the laboratory, the tissues without the shell were worked, each sample was stored in a refrigerated/frozen ziplox bag with the meat of 20 individuals corresponding to replica 1, 2 and 3 of each sampling.

Please add more details….” This comment wasn't addressed, neither in the reply nor in the revised ms. 

Reply: We improved the paragraph, including the required information in the manuscript. We appreciate your helpful suggestions.

Line 121ff: Please comment on using the calculation of yield instead of condition index (soft body dry mass*100/shell dry mass) which is more common. In any case, data sets should be compared with literature data to increase the reliability of the results.

Reply: Samples were processed in the field, where 20 individuals were randomly selected for each sample, measured and weighed wet, shells were immediately removed and the soft bodies of the 20 individuals were grouped in a properly labeled bag. The bags were stored cold for transport to the laboratory. In the laboratory, the samples were frozen, freeze-dried, macerated and grouped by bag, kept frozen until analysis for proximal and fatty acid content.

The index was calculated according to Hickmann and Illingworth (1980).

These authors compared different condition index formulas that are weight/weight ratios and that reflect changes in body proportions in bivalve populations.

CIwet = [wet meat volumen/(whole volumen - Shell volumen)] * 100

CIvolume = [dry meat wt / (whole volumen - shell volumen)] * 100

CIweight = [dry meat wt/(whole wt - shell wt)] * 100

CIcommercial = [Meat wet wt / Total wet weight] * 100

They found that CIcommercial correlated 92.7% with CIwet, 90.4% with CIvolume and 90.9% with CIweight.

According to Hickmann and Illingworth (1980), although CIweight is the most advisable for reliable comparisons, it is possible to standardize the measurement of wet weight in an index such as CIcommercial to minimize errors by always adopting the same weighing routines, which is what was applied in this study.

Line 133: Please clarify how samples were treated prior to analysis to ensure homogeneity, as parameters were determined in whole animal extracts.

Reply: For each sample of 20 individuals the tissues were pooled and freeze dried in a Savant freeze dryer (-80°C), then they were ground to homogenize and stored at -41°C until the respective biochemical analysis.

This is an important information that the tissue of 20 individuals was pooled prior to analyses and must be given in the ms. 

Reply: Thank you for the clarification. This information has been included in the manuscript.

Line 133: Unclear, please add more information how crude protein was calculated.

Reply: In nutrition, the total nitrogen content of the sample was converted into a value called "crude protein" by multiplying it by 6.25, a constant based on the conventional assumption that any protein is composed of 16% nitrogen.

Again, this is an information that should be given in the ms. 

Reply: Thank you for the clarification. This information has also been included in the manuscript.

Line 146f: Repetition, see line 131f.

Reply: corrected

Sorry, the repetition is still in the revised version. 

Reply: This repetition has been removed. Thanks for the clarification

First revision, R1 (January 2022)

REVIEWERS #1 and #2 COMMENTS:

Review Comments to the Author

Reviewer #1: The paper by A. Valenzuela et al. describes interesting data on the proximal composition and fatty acid content of three mollusc species from Chile, that can contribute to the literature already existing on bivalve molluscs.

The Authors also tried to assess differences existing between species or even between the same species growing in relatively close locations.

On the other hand, several aspects of the paper need to be improved, especially in terms of description of experimental procedures, statistical analysis and results.

A list of suggestions/comments that could help the Authors during the revision stage is thus provided below, following the order of appearance in the manuscript.

Materials and methods

The title of the first sub-section (Sampling) of this section should be integrated to clarify that the sub-section deals also with the determination of proximal composition and fatty acid profile of molluscs.

Reply: Done

Lines 122-129: the use of the same symbol, WW, first for the calculated wet weight (lines 125-126) and then for the experimental wet weight (lines 128-129) can be misleading. A different subscript (i.e., calc and exp, according to the case) should be added to the WW symbol to avoid any misunderstanding. Wr would thus be expressed as WWexp/WWcalc.

Reply: The WW values cited between lines 122-129 are any WW obtained experimentally to apply the index Wr, that allows to determine if a WW sampled in an individual of species A relates to the expected WLR for species A with a values inferior, similar or superior to 1; ie, there are no calculated WW values, only experimental values in all cases

Line 133. Details should be provided about the use of the N x 6.25 rule. This can not immediately understandable for all readers.

Reply: In nutrition, the total nitrogen content of the sample is converted into a value called "crude protein" by multiplying it by 6.25, a constant based on the conventional assumption that any protein is composed of 16% nitrogen.

Line 137 – 138. Which detector was used after GC separations? I guess that it was a FID but this detail has to be provided.

Reply: Flame ionization detector (FID) Thermo Scientific was used

Lines 146-150. The sentence was already reported at lines 131-135.

Reply: corrected

Lines 151-152. Is there any previous reference about this approach to estimate the carbohydrate content?

Reply: the methods of substraction of total carbohydrate from proximal composition as cited by Metzger LE and Nielsen SS, 2019. Chapter 3 Nutrition Labelling. In: Food Analysis. Fifth edition, SS Nielsen (ed), Food Science Text Series, Springer, Cham, Switzerland, pp 35-44

Lines 157, 158, and 160-162. The “g dry weight” expression should be put between parentheses and then the -1 exponent should be placed outside; as an alternative, the word “per” could be placed before “g dry weight”.

Reply: done

Lines 173-176. The type of data used as original variables for PCA must be cited.

Reply: The data used for PCA analysis were the values of each replicate for: every fatty acid, every relationship between fatty acids, every performance indicator, and every proximal content. Finally, only those variables most correlated with the first two components were selected for final PCA analysis.

Results

Lines 188, 190, 193 and 196: the meaning of the H and F symbols needs to be clarified. Since the Authors claimed that normality and homoscedasticity tests were performed om data, the results of those tests should also be described.

Reply: done

Line 213. Fig2 should be referred to as Fig2a.

Reply: done

Figure 2. I would strongly suggest adding also loading plots to the figure, to help the readers in understanding how the variables influenced the differences observed between samples in the score plot. It is very complicated to switch to the Supplementary Material to find this information using Figure S1.

Reply: Figure 2a and b was improved and replaced with data from the supplementary material. Therefore, the supplementary material was eliminated, and its data were incorporated along with the variables in Figure 2, for a better understanding of the results.

Tables 1-4. It seems that the results of tests for the comparison between means were not reported for all data, as significant differences could be supposed to be present sometimes (based on numerical values) but superscripts indicating the outcome of those tests were missing, thus suggesting that no significant difference was observed. Is this correct?

I would suggest using a consistent number of significant figures to express standard deviations (for example two) and then round off mean and standard deviation values accordingly.

The number of replicates should be clearly indicated in the tables captions.

Reply: done.

Line 309. I would suggest using the word “quadrant” or “semi-plane”, according to the case, instead of “hemisphere”.

Reply: done.

Lines 306-311: it would be better to use the term “principal” instead of “main”, since PCA results are described. I would be very careful in writing that Chilean oysters from Pullinque and Quempillen made up different groups in the score plot. Indeed, a remarkable variability was observed for the samples collected in Pullinque and at least one of them was very close to the samples from Quempillen.

Reply: done.

Figure S1. What is the meaning of the colour code (and then of the legend) adopted in the figure?

Reply: In blue (active) it refers to the active variables that influence the principal components and in red (suppl) to the supplementary variables that have no influence on the analysis of the principal components, but do help to interpret the results. However, this graph was corrected and the supplementary material was removed, Figure 2 and Figure S1 were merged for a better understanding of the results.

Figure 3. The figure does not show daily consumptions, as stated in the caption, but the amounts of the two fatty acids referred to 100 g of meat of each mollusc. Please clarify this issue. The number of replicates used to calculate error bars should be indicated for each set of data.

Reply: Figure 3. Potential intake of EPA + DHA in a 100 g edible portion from Crassostrea gigas, Ostrea chilensis and Mytilus chilensis.

Lines 375. Why were just those two values reported for the n-3/n-6 ratio? If the Authors meant to indicate the entire interval observed for those ratios in all the analyzed species, then the minimum value would be 5.15, found for Chilean oysters from Pullinque. A clarification is needed.

Reply: corrected

Lines 412-413. This sentence seems to be conceptually linked to the previous one, yet it concerns animal genetic improvement, so how can it be related to the sentence in which the focus was on the locality? Please clarify.

Reply: corrected

Conclusions. I would suggest rephrasing this section, focusing on more specific aspects of the work, like the comparison between oysters and mussels in terms of key components and the subtle effects that the location seemed to have on the nutrient content of the considered shellfish.

Reply: Reworded this section as suggested by reviewer.

Reviewer #2: Present study determined the proximal biochemical composition and the fatty acid profile in the Chilean oyster (Ostrea chilensis), the Pacific oyster (Crassostrea gigas) and the Chilean mussel (Mytilus chilensis), to perform an intra and interspecific comparison. To my understanding the interspecific comparison (O. chilensis, C. gigas and M. chilensis) was performed on samples that were all collected from Hueihue in 2018, and the intraspecific comparison was meant on O. chilensis that were sampled from two locations (Quempillen vs. Pullinque) in 2017. Additionally, to the results of O. chilensis (2017) the authors present results of C. gigas (sampled from Faro Corona in 2017). However, this is irritating as C. gigas is a different species and the results do not fit to the “intraspecific comparison”.

Besides minor grammatical inaccuracies, the ms is well written and the results seem to be reliable supporting the conclusion that the investigated shellfish species can be a “sustainable option for a balanced and healthy human diet”. However, the finding that oysters and mussels are healthy food components in human diet is not new and the study will benefit from a clearer (hypothesis-driven) research question to more strongly emphasize its relevance to the scientific community. In addition, the ms can be improved by a more detailed description of the M&M part and an in-depth discussion.

Major comments:

1) The authors aim to “nutritionally characterize the food quality” of two oyster and one mussel species from aquaculture in Southern Chile. To improve the study, the results presented (content of lipids, protein etc. as well as fatty acid composition) need to be put in context with literature data and discussed appropriately; e.g. Do the results differ from bivalves obtained from European or US aquaculture facilities? If so, why? Investigations were conducted on bivalves collected during the winter season, and the data could be discussed/compared with data from bivalves collected during spring/summer.

Reply: This comment has been considered in the discussion to deepen the comparison between the three species of bivalves.

2) Please highlight why the different locations were chosen for the intraspecific comparison. In what respect do they differ from each other? The intraspecific comparison aspect should be more clearly elaborated in the discussion. In its present form (under the heading “Differences between populations of oysters”) it is a bit confusing as the dataset of C. gigas is also discussed. Please comment on your decision to add the data of C. gigas (obtained in 2017) to the results part of the intraspecific comparison (line 243f).

Reply: The locations chosen in this study for the comparison between oyster species were selected because Quempillén and Pullinque correspond to two areas of natural banks of O. chilensis on Chiloé Island. The town of Quempillén is an estuarine zone with high salinity fluctuations. Instead, the town of Pullinque is a bay in the Gulf of Quetalmahue with depths of 7 to 8 meters. The species C. gigas was added because it is the second most important commercially important oyster species in Chile and because it also has natural banks on Chiloé Island.

3) With respect to the results in Table 1 showing highest proximal composition values in protein content (42 – 52% dry weight) and lower values in lipid (14.5 – 19.5%) and carbohydrates (11.5 to 20.9%), the statement “shellfish sampled in winter were characterized by high lipid content along with low protein and carbohydrates contents (line 36f, and 431f) is irritating. Please clarify.

Reply: The paragraphs mentioned in lines 36f and 431f were corrected.

Specific comments:

Line 26/27: Please clarify the definitions of equal contribution of the authors - ¶ vs. &.

Reply: the authors' equal contribution to:

 ¶ = all of these authors contributed to study design, data collection and analysis, decision to publish, or manuscript preparation.

&= all these authors contributed to the experimental and field work.

Line 83: Change into “On the one hand…”

Reply: done.

Line 107f: Sampling of animals: Please clarify the sentence "3 samples of 20 oysters" as it is 60 animals in total, isn't it? Please add more details about the environmental parameters at the time of collection (water temperature, salinity...) and transport of the animals (duration, transport on ice/water...). Were the animals kept in institutional aquarium systems prior to dissection? If yes, please add details on husbandry.

Reply: For each locality, 60 sampled specimens were collected from cultivation areas, where 3 samples (replicates) of 20 individuals were randomly made, except for C. gigas, which were fewer specimens (40 individuals). Then, in the laboratory, the tissues without the shell were worked, each sample was stored in a refrigerated/frozen ziplox bag with the meat of 20 individuals corresponding to replica 1, 2 and 3 of each sampling.

Line 121ff: Please comment on using the calculation of yield instead of condition index (soft body dry mass*100/shell dry mass) which is more common. In any case, data sets should be compared with literature data to increase the reliability of the results.

Reply: The samples were processed in the field, where the wet weight was directly measured. In our laboratory, the samples were lyophilized to be considered for proximal content and fatty acid content. The work of Sing & Ransangan (2019) measure yield according to wet weight.

Sing, O. F., & Ransangan, J. (2019). Effect of physicochemical parameters and phytoplankton composition on growth performance of green mussel (Perna viridis) in Ambong Bay and Marudu Bay, Sabah, Malaysia. Journal of Fisheries and Environment, 43(1), 50-68.

Line 122: Please add the underlying data for the calculation of the allometric relationship of WW. The cited reference (#30) is for Wr for fish. Is it also used for shellfish?

Reply: Now the underlying data suggested is: 

The relative weight index [30] is based on the length -weight relationships (LWRs). The total wet weight in grams (WW), and shell length in milimeters (L) obtained from each individual sampled during the study were used. The wet weights were fit using a power regression as a function of shell lenght to obtain WW = aLb , where a is the intercept (condition factor) and b is the slope (relative growth) following Sousa et al (2020 a, b). The relationships were WW = 0.021*L2.16 for M. chilensis with 60 individuals (R2= 0.48), WW = 0.0004*L2.77 for O. chilensis with 180 individuals (R2= 0.51), and WW = 0.079*L1.48 for C. gigas with 100 individuals (R2= 0.67).

The use of WLRs are commonly used for shellfish see Sousa et al 2020a, b. The WLRs are used to compare different ecological, geographical, reproductive, etc., conditions for some particular species, and the comparisons is about the exponent of WLRs. 

So the Wr is only a ratio WW/WLR to show if the animals are more slim or fat respect to an average condition represented by WLR of the species in different geographical areas, or growth conditions, or etc, Therefore it is applicable to shellfish, the same as WLRs

Line 133: Please clarify how samples were treated prior to analysis to ensure homogeneity, as parameters were determined in whole animal extracts.

Reply: For each sample of 20 individuals the tissues were pooled and freeze dried in a Savant freeze dryer (-80°C), then they were ground to homogenize and saved at -41°C until the respective biochemical analysis.

Line 133: Unclear, please add more information how crude protein was calculated.

Reply: In nutrition, the total nitrogen content of the sample is converted into a value called "crude protein" by multiplying it by 6.25, a constant based on the conventional assumption that any protein is composed of 16% nitrogen.

Line 146f: Repetition, see line 131f.

Reply: corrected

Line 161: Please provide more details concerning the equation parameters “12g” and “15g”.

Reply: correspond to the molecular weights of carbon and nitrogen. The nitrogen value was corrected to 14.

Line 163: Consider rephrasing “performance”. If it is a common term in aquaculture, please provide a citation for the calculation.

Reply: CI commercial = wet meat weight x 100 /whole wet weight (live) = Meat wet weight x 100/ Total wet weight following Hickman and Illingworth (1980)

Line 165f: Statistics: Missing info of p level.

Reply: p< 0.05

Line 191: Revise sentence “higher compared to …”. Please check the whole ms regarding this aspect. Furthermore, using the mean values of lipid content (Table 1) the values (9 to 25%) do not fit. Please clarify.

Reply: corrected

Line 202: Table 1 and all other Tables: Please add experimental number. Energy is expressed as MJ per kg dry meat, isn’t it? Please specify that “different” letters indicate differences.

Reply: done

Line 206: see comments to line 191.

Reply: done

Line 317f: What about the EDA+DHA concentration of nearly 600mg/100g ww meat determined in C. gigas samples collected in 2017?

Reply: For 2017 samples, an amount of 67 g of Chilean oyster from Quempillén sampling site or 92 g of Chilean oyster from Pullinque or 44 g of Pacific oyster are needed to achieve daily requirement (Fig.3).

Line 368f: This is a repetition, see line 317f.

Reply: corrected

Line 375: Please clarify: 7.13 “y” 12.86.

Reply: corrected.

Line 376: Please add more details/information. The reader wonders, so what?

Reply: Reworded this paragraph to be more consistent.

---

## [Editor Report · Decision Letter 3]

21 Jun 2022

Proximal and fatty acid analysis in Ostrea chilensis, Crassostrea gigas and Mytilus chilensis (Bivalvia: Mollusca) from Southern Chile.

PONE-D-21-28284R3

Dear Dr. Valenzuela,

We’re pleased to inform you that your manuscript has been judged scientifically suitable for publication and will be formally accepted for publication once it meets all outstanding technical requirements.

Kind regards,

Frank Melzner

Academic Editor

PLOS ONE

---

## [Editor Report · Acceptance letter]

23 Jun 2022

PONE-D-21-28284R3 

Proximal and fatty acid analysis in *Ostrea chilensis, Crassostrea gigas* and *Mytilus chilensis* (Bivalvia: Mollusca) from Southern Chile 

Dear Dr. Valenzuela:

I'm pleased to inform you that your manuscript has been deemed suitable for publication in PLOS ONE. Congratulations! Your manuscript is now with our production department. 

Kind regards, 

on behalf of

Dr. Frank Melzner 

Academic Editor

PLOS ONE